

# Atmospheric ice-nucleating particles in the Eastern Mediterranean and the potential influence of fertile soils

Mark D. Tarn[1], Bethany V. Wyld[1], Naama Reicher[2], Matan Alayof[2], Daniella Gat[2], Alberto Sanchez-Marroquin[1], Sebastien N. F. Sikora[1], Alexander D. Harrison[1], Yinon Rudich[2], Benjamin J. Murray[1]

[1]School of Earth and Environment, University of Leeds, Leeds, LS2 9JT, United Kingdom
[2]Department of Earth and Planetary Sciences, Weizmann Institute of Science, Rehovot 76100, Israel

*Correspondence to*: Mark D. Tarn (m.d.tarn@leeds.ac.uk), Benjamin J. Murray (b.j.murray@leeds.ac.uk)

**Abstract.** While the atmosphere in the Eastern Mediterranean is part of the dust belt, it encounters air masses from Europe, the Mediterranean Sea, and the Sahara and Arabian Deserts that bring with them a whole host of potential dust and bioaerosol

compositions and concentrations via long-range transport. The consequential changes in the populations of ice-nucleating particles (INPs), aerosols that influence weather and climate by the triggering of freezing in supercooled cloud water droplets, including in the convective cloud systems in the region, are not so well understood beyond the influence of desert dust storms in increasing INP concentrations. Here, we undertook an intensive INP measurement campaign in Israel to monitor changes in concentrations and activity from four major air masses, including the potential for activity from biological INPs. Our

findings show that the INP activity in the region is likely dominated by the K-feldspar mineral content, with southwesterly air masses from the Sahara Desert and easterly air masses from the Arabian Deserts markedly increasing both aerosol and INP concentrations. Most intriguingly, air masses that passed over the Fertile Crescent, a crescent-shaped region of fertile soils, agriculture, wetlands, and marshes stretching from the Nile Delta and over northern Syria to eastern Iraq, brought high INP concentrations with strong indicators of biological activity. These results suggest that the Fertile Crescent could be a sporadic

source of high-temperature biological ice-nucleating activity across the region that could periodically dominate the otherwise K-feldspar-controlled INP environment and warrants further exploration in future studies in the region. This is particularly true given the ongoing desertification of the Fertile Crescent that could reveal further sources of dust and fertile soil-based INPs in the future.

## 1 Introduction

Clouds play an important role in weather and climate due to precipitation and their radiative properties, but their representation in cloud-feedback and climate models retains a high degree of uncertainty due to poor representation of a number of dynamic and microphysical processes, including cloud-aerosol interactions (Ceppi et al., 2017; Storelvmo, 2017). A rare class of aerosol particles known as ice-nucleating particles (INPs) are able to catalyse the formation of ice in supercooled cloud water droplets at temperatures higher than around −35 °C (Murray et al., 2012;





Hoose and Möhler, 2012; Kanji et al., 2017), below which cloud droplets can freeze homogeneously (Herbert et al., 2015; Rosenfeld and Woodley, 2000).

The presence of INPs can thus alter the ratio of water to ice in mixed-phase clouds and influence their albedo and lifetime (Lohmann and Feichter, 2005). Therefore, an accurate representation of INPs in global models is important, particularly in a changing climate (Murray et al., 2021), but this is currently lacking (Kanji et al., 2017;

Vergara-Temprado et al., 2017) since INPs are not only rare but also highly variable globally in terms of their activity, concentrations, transport, processing, sources and sinks.

To better characterise atmospheric INPs, it is necessary to undertake field campaigns that collect essential and seasonal data on INP properties from around the globe and to determine the origins of those INPs and their effects on clouds. The Eastern Mediterranean is a fascinating location in terms of the varying air mass trajectories that reach

it and the potential for vastly different INP communities associated with each, which could influence the clouds in the region, e.g. deep convective cloud systems (Funatsu et al., 2007). A large proportion of air flowing into deep convective clouds can come directly from the boundary layer, hence boundary layer aerosol are of direct relevance for the development of deep convective clouds (Hoffmann et al., 2015; Hawker et al., 2021). Dayan (1986) studied five-day back trajectories from Israel over a five-year period and identified five major source regions of air masses:

(i) northwest Europe, crossing the Mediterranean Sea, throughout the whole year, (ii) north-easterly continental flow originating in Eastern Europe during the summer season, (iii) infrequent south-easterly flow from the Syrian and Arabian Deserts that occurred mainly during Autumn (fall), (iv) south-westerly flow along the North African coast most frequent during late winter and spring, and (v) south-southwesterly flow from inland North Africa that mostly occurred during winter and spring.

One of the major features of the region are the dust events associated with the western Sahara Desert (Dayan et al., 1991; Ganor, 1994, 1991; Ganor and Mamane, 1982; Yaalon and Ganor, 1979; Mamane et al., 1980; Athanasopoulou et al., 2016) and the Syrian and Arabian Deserts. Transported dust can have residence times of hours to weeks (Ganor, 1991, 1994; Kubilay et al., 2000), and can cover heights of up to 6 km above mean sea level (a. m. s. l.) during transport (Israelevich et al., 2002). Dayan et al. (1991) found that Saharan events usually have high dust

loadings and last for 2-4 days. Conversely, those from the Arabian Desert last for shorter periods (~1-2 days) and are restricted to a more shallow atmospheric transport layer (up to ~2000 m a. m. s. l.) that may prevent their passing much beyond the Eastern Mediterranean.

Saharan dust events are important in terms of INP population and sources in the Eastern Mediterranean (Ganor, 1994), particularly as the number of dust days caused by African dust each year increased decade-by-decade

until around 2010 (Ganor et al., 2010), although this trend has recently reversed (Nissenbaum et al., 2023). In addition to dust, westerly air flows can bring high levels of sea salt from the Mediterranean Sea (Levin and Lindberg, 1979; Foner and Ganor, 1992) and biogenic sulphate aerosols (Ganor et al., 2000). Dust storms that pass over the Mediterranean Sea can contain internally mixed dust and sea salt (Levin et al., 2005).



65   There have been a handful of INP measurement campaigns in the Eastern Mediterranean region, specifically around Israel, and it is unsurprising given the importance of Saharan dust that several studies have either focussed on the effect of Saharan dust events or have drawn conclusions from them. Levi and Rosenfeld (1996) performed ground-level filter sampling of aerosols with INP analysis using a thermal diffusion chamber, finding that INP counts more than doubled during "dust days" compared to "clean days", and further that a major fraction of the dust that they collected in rainwater samples originated from the Sahara Desert.

70   This conclusion was backed up by the more recent work of Ardon-Dryer and Levin (Ardon-Dryer and Levin, 2014), who performed ground-level (60 m a. m. s. l.) aerosol filter sampling with immersion mode INP analysis achieved using the FRIDGE-TAU (Frankfurt Ice-nuclei Deposition freezing Experiment, the Tel Aviv University version) droplet assay. Throughout the campaign the researchers saw southwesterly (Saharan dust) or northwesterly (European) air masses, with the INP data again showing a doubling of INP activity during the "dust days" of the former case and the "clean days" of the latter. Further, analyses during the annual Lag BaOmer bonfire festival event

showed that, despite an increase in total aerosol concentrations there was no associated increase in INPs, which is consistent with studies during similar bonfire and firework events in the UK (Adams et al., 2020).

  Boose et al. (Boose et al., 2016) performed INP analysis of a series of Saharan and non-Saharan transported airborne dust samples from a number of locations, and surface samples from other locations, including sieved and

milled surface samples from Israel. Surprisingly, the authors found that the surface-collected Israel dust demonstrated some of the lowest ice-nucleating activity amongst all of the other samples.

  Reicher et al. (Reicher et al., 2018; Reicher et al., 2019) studied the ice-nucleating ability of size-resolved aerosol collected during dust events in Israel, analysed using the Weizmann Supercooled Droplets Observation on a Microarray (WISDOM) microfluidic instrument. Their results showed that ice-nucleating ability increased with

particle size and concentration. The activity of supermicron particles was similar for different dust events, suggesting common mineral species controlling the nucleation.

  Gagin (1975) performed aircraft and ground-based filter sampling of aerosols with INP analysis via a thermal diffusion ice nucleus counter to study winter cumulus clouds over Israel, finding that INP concentrations roughly approximated the measured ice crystal concentrations within about one order of magnitude.

90   Campaigns have taken place in Cyprus in the Eastern Mediterranean, northwest of Israel. Schrod et al. (2017), during the INUIT-BACCHUS-ACTRIS campaign, studied vertical profiles over Cyprus using unmanned aircraft systems (UASs) with electrostatic precipitation-based sampling to collect INPs for analysis by FRIDGE. Ice-nucleating activity was dominated by Saharan dust in the lower and middle troposphere, with concentrations of an order of magnitude higher than at ground level that may have comprised weak marine and terrestrial sources in

Cyprus.

  As part of the same INUIT-BACCHUS-ACTRIS campaign, Marinou et al. (2019) used ground-based and spaceborne lidar observations combined with INP parameterisations (DeMott et al., 2010; DeMott et al., 2015) to





develop a new method for the vertical profiling of INPs. Validation was achieved using the vertical profile data of Schrod et al. (2017) obtained using UASs and FRIDGE.

Gong et al. (2019) undertook a ground-based filter sampling campaign in Cyprus with INP analysis achieved using the Leipzig Ice Nucleation Array (LINA) and Ice Nucleation SpEctrometer of the Karlsruhe Institute of Technology (INSEKT) instruments. Air masses were divided into "land" and "ocean" sectors, with no significant differences found between their INP concentrations at temperatures lower than −15 °C, implying that this INP population was dominated by long range transported aerosol. However, a few samples showed elevated activity at

temperatures greater than −15 °C, which suggested a population of biological INPs speculated to originate in Cyprus. As a result, the authors proposed that the standard methods of parameterising the INP activities (e.g. for desert dusts), such as the ice-active surface site density ($n_s(T)$), are unsuitable if the aerosol particle composition is unknown.

During a ship cruise in the western and central regions of the Mediterranean Sea, Trueblood et al. (2021) performed sampled sea-spray aerosol (SSA) and the sea surface microlayer (SML) marine and analysed their INP

content using a Dynamic Filter Processing Chamber (DFPC). SML results showed lower INP concentrations than expected from the literature and was presumed to be due to the oligotrophic nature of the Mediterranean Sea. A dust wet deposition event demonstrated that, while INP concentrations in the SML increased by an order of magnitude almost immediately, the concentrations in the SSA took 3 days to increase.

Prodi et al. (1983) performed aerosol sampling and INP analysis via a static diffusion chamber at −16 °C

during a ship campaign in 1979 that crossed the Mediterranean, Red, and Arabian Sea. The highest INP concentrations were measured in the southern section of the Red Sea, and the authors noted a correlation with the mass concentration of mineral aerosol apart from then SSA production was high. However, we note several other instances in their data where the INP data increased much higher relatively speaking than the mineral content did, suggesting that other non-mineral INPs, e.g. biological INPs, may have been present.

Beall et al. (2022) undertook a more recent cruise around the Arabian Peninsula in which they measured INP concentrations in aerosol and seawater using the Scripps Institution of Oceanography automated ice spectrometer (SIO-AIS) droplet assay. Several days' of data were collected in the Red Sea with air masses from the Mediterranean, the Nile Delta and the Sinai Peninsula that each contained some degree of dust loading. Following heating and peroxide treatment of the samples, the authors noted that agricultural soil dust from the fertile Nile Delta region

exhibited greater heat sensitivity that is indicative of biological INP content. Thus, the authors suggested that INP activity at temperatures below around −15 °C, while the activity at warmer temperatures was likely due to fertile soils from the agriculturally important Nile Delta region, which comprises part of the Fertile Crescent region that arcs from the Nile Delta through Israel, Lebanon, and northern Syria and southern Turkey, then down through northern and eastern Iraq and western Iran. As a consequence of this and in noting that aerosol aging can have an effect on INP

activity, they, similar to Gong et al. (2019), proposed that mineral dust parameterisations alone may not suitable for



the representation of regional INP activity when the INP population contains unknown INP populations that are not dominated by the mineral dust content.

Roesch et al. (2021) studied cloud condensation nuclei (CCN) and INPs from three locations on the Saudi Arabian peninsula, finding that surface samples from the east coast were ice nucleation active at warmer temperatures than those in the northeastern and central regions of the country, highlighting that INPs from a relatively small region could have very different properties. The authors also found that INP activity was similar to that of Saharan dust.

To better understand INP concentrations and activity in the Eastern Mediterranean region beyond dust events, we undertook an intensive two-week field campaign in Rehovot, Israel, in October-November 2018 where we collected ground-based filter samples for INP analysis by immersion mode droplet freezing assays. Given the variety of air masses received by this location, we hoped to determine whether some sources could potentially influence local INP populations over the typical dust events.

During this campaign we experienced four major air masses that we were able to study: (i) a southwesterly dust event from the Sahara Desert that passed over the fertile Nile Delta region, (ii) a northwesterly air mass from Europe, (iii) a prolonged easterly event from the Arabian Desert that contained air with both high- and low-loadings of dust, and (iv) an air mass that passed over the agriculturally important northern Fertile Crescent region before sweeping eastward to the sampling site.

## 2 Experimental

### 2.1 Aerosol sampling and preparation of particle suspensions

Atmospheric aerosol samples were collected over 10 days, from 25/10/18 to 03/11/18 (DD/MM/YY), from the roof of the Department of Earth and Planetary Sciences building at the Weizmann Institute of Science in Rehovot, Israel (31.9° N, 34.8° E, ~80 m a. m. s. l.). Aerosol sampling was performed using two different methodologies: (1) the collection of aerosol particles onto filters, and (2) the sampling of aerosol particles into water using an impinger. Aqueous suspensions of particles were prepared in both instances using purified water (18.2 MΩ cm at 25 °C, 0.22 µm filtered) from a Thermo Scientific™ Barnstead™ GenPure™ water purification system.

Filter-based sampling for INP analysis was performed using our previously described technique (O'Sullivan et al., 2018). Two BGI PQ100 Air Sampling Systems (Mesa Laboratories, Inc., Lakewood, CO, USA) with $PM_{10}$ (particulate matter of ≤10 µm diameter) inlet heads were purchased from Air Monitors (Tewkesbury, UK) and were employed in order to draw air through porous filters onto which aerosol particles were adsorbed. The BGI PQ100 is used as an Environmental Protection Agency (EPA) Reference Method for $PM_{10}$ (designation no. RFPS-1298-124). $PM_{10}$ aerosol were sampled onto Whatman® Nuclepore™ track-etched membrane polycarbonate filters (1.0 µm pore size, 47 mm diameter filters, purchased from Sigma-Aldrich, Dorset, UK) at a flow rate of 16.67 L min⁻¹ (1 m³ h⁻¹). While the filters had 1.0 µm pore sizes, studies have demonstrated that membrane filters are able to capture particles





much smaller than the given pore size (Soo et al., 2016; Lindsley, 2016); 1.0 µm pore size polycarbonate filters have been shown to have collection efficiencies of 50-90 % for particle sizes of 0.01-0.2 µm, and 90-100 % for particle
sizes larger than 0.2 µm (Burton et al., 2006; Lindsley, 2016).

Filter-based sampling for INP analysis was performed using two main strategies: (i) the collection of samples for 3 hours every morning and 3 hours every afternoon (yielding 3,000 L of sampled air for each sample), and (ii) the collection of samples for 24 h from midday to midday (24,000 L of sampled air). The 3 h samples were collected using one PQ100 sampler and the 24 h samples using a second PQ100 sampler. Collection of the 3 h samples was
intended to provide a relatively short time resolution of the INP measurements, while the 24 h samples were originally collected to achieve a large volume of sampled air since it was not immediately clear how much air would need to be sampled to obtain an INP signal. However, the sampler used for the 24 h samples failed during collection of the fourth sample between 28th-29th October after 17.5 h, and no further 24 h samples were collected from that point.

In one experiment during the afternoon of 29th October 2018, the two samplers were run side-by-side for 3
h, with one collecting $PM_{10}$ aerosol onto a filter as usual and the other collecting $PM_1$ aerosol simultaneously onto a second filter via the use of a $PM_1$ adaptor (SCC 2.229 Cyclone, Mesa Labs) inserted into the sampler. Two handling blanks were also performed for the filter-based sampling by attaching a HEPA (high-efficiency particulate air) filter to the BGI PQ100 sampler and pulling filtered air through a track-etched membrane filter. One handling blank filter was sampled for 1 h during the early stages of the campaign, and another filter sampled for 3 h during the latter stages.

Details of the sampling times and volumes of air sampled for each filter collected are provided in Table S1 in the Supplementary Information (SI). Filters collected for INP analysis were removed from the BGI PQ100 sampling system and immediately inserted into a 50 mL centrifuge tube (Sarstedt Ltd., Leicester, UK) using tweezers, then 4 mL of purified water added via pipette. The tube was shaken vigorously for several seconds before being vortexed on a vortex mixer (Labnet VX100) for 5 min, allowing the adsorbed aerosol particles to be washed off the
filter and into suspension ready for analysis via a droplet freezing assay.

Impinger samples were collected during the afternoon for 7 days of the campaign using a Bertin Technologies Coriolis® Micro air sampler, purchased from Air Monitors. This impinger measures particles larger than 0.5 µm, with a collection efficiency, $D_{50}$ (i.e. the particle size at which the collection efficiency is 50 %), of <0.5 µm). Impingers have previously been successfully employed by several research groups for INP sampling and analysis,
(Šantl-Temkiv et al., 2017; Garcia et al., 2012) including the use of a Coriolis Micro for the detection of ice-nucleating proteins by genetic sequencing (Els et al., 2019). Here, ambient air was aspirated into a cone filled with 10 mL of purified water, with aerosol particles that entered the sampler inlet being centrifuged to the wall of the cone and extracted from the air into the water. Sampling was performed in multiple 10 min intervals (the maximum runtime of the sampler), at a flow rate of either 100 L min$^{-1}$ or 300 L min$^{-1}$, until 6,000 L of air had been sampled into the water-
filled cone. A lid was screwed onto the cone once it had been removed from the sampler to prevent contamination.

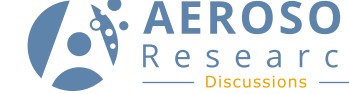

Purified water was added to the cone following each sampling interval due to evaporational losses during sampling, and the final volume of water in the cone (~5-10 mL) was determined by weight at the end of the collection period. A handling blank was performed for the impinger samples by filling a sampling cone with purified water, attaching it to the impinger, and then removing it and screwing a lid onto it ready for droplet freezing assay analysis of the
water. It was not possible at the time to pull HEPA-filtered air through the sampler for the handling blank due to the awkward shape of the inlet. Details of the sampling times and final water volumes are provided in Table S1 in the ESI. The water from the cone was analysed directly via a droplet freezing assay.

**2.2 Heat treatment of aerosol suspensions**

An aliquot of every aqueous suspension from the filter samples and the impinger samples was subjected to a heat
treatment test. Heat treatments are used to test for the presence of heat-labile ice-nucleating materials (O'Sullivan et al., 2018; Christner et al., 2008b; Christner et al., 2008a; Daily et al., 2022), which are generally equated to ice-nucleating proteins that suggest the presence of biogenic INPs (e.g. bacteria or fungi). Such tests can only be used to indicate of the possibility of biogenic INPs being present but cannot alone provide definitive evidence. Since some minerals can also lose their activity upon heating, the assumption that a loss of activity may be due to ice-nucleating
proteins is considered to only be viable if the main mineral dust component is microcline K-feldspar, which does not lose activity following heat treatment (Daily et al., 2022).

The aerosol compositions of dust events in Israel mainly compromise calcite ($59 \pm 15$ %), quartz ($23 \pm 7$ %), dolomite ($11 \pm 8$ %), feldspars ($5 \pm 3$ %), halite ($2 \pm 1$ %) and traces of clay minerals (Ganor and Mamane, 1982; Foner and Ganor, 1992; Ganor, 1975). While the carbonates (calcite and dolomite) and quartz dominate the
composition, it is the feldspars that are thought to be the most important factor for the nucleation of ice in mixed-phase clouds (Atkinson et al., 2013; Kiselev et al., 2017). While different feldspars have vastly different ice-nucleating activities (Harrison et al., 2016), microcline potassium (K)-rich feldspar exhibits much greater activity than quartz or Na-Ca feldspars (plagioclase series) (Harrison et al., 2019).

As a consequence, the presence of K-feldspar, even at concentrations of around only 1 % (Harrison et al.,
2022), is expected to dominate the ice-nucleating ability of airborne desert dust. Boose et al. (2016) analysed sieved and milled surface dust from Israel and found K-feldspar contents of $1.7 \pm 0.3$ % and $1.4 \pm 0.3$ %, respectively, and agreed with Ganor et al. (Ganor and Mamane, 1982; Foner and Ganor, 1992; Ganor, 1975) that the samples comprised mostly of carbonates. Based on this, we assume here that microcline K-feldspar is the dominant component of the mineral dust in Israel in terms of ice-nucleating activity, and so the heat test for biogenic INPs remains viable.
Heat treatment was performed by adding 1 mL of aqueous suspension to a sealed glass vial and heating at 95 °C for 30 min in a bath of boiling water, consistent with the recommendations of Daily et al. (2022). The sample was then allowed to cool to room temperature prior to analysis via a droplet freezing assay for comparison to the untreated



sample. A control test was also performed by performing droplet freezing assays on an aliquot of purified water before and after being subjected to heat treatment.

**2.3 Droplet freezing assays for INP analysis using the µL-NIPI**

All aqueous suspensions from the filter-based samples (3 h and 24 h collection periods) and the impinger-based samples were analysed using the Microlitre Nucleation by Immersed Particle Instrument (µL-NIPI) (Whale et al., 2015) immediately following their collection and preparation. This involved setting the droplet freezing apparatus up in the laboratories of the Weizman Institute of Science. The advantage of analysing the samples immediately

following collection was that we reduced the possibility of changes in the INP population that have been observed on storage and transport of filter samples (Beall et al., 2020).

Hydrophobic glass slides (22 mm Ø × 0.22 mm siliconised glass, Hampton Research, Aliso Viejo, CA, USA) were washed with ethanol and purified water, dried with dry nitrogen gas, and placed atop the cold plate of a Stirling engine-based cold stage (EF600, Grant Asymptote Ltd., Cambridge, UK).

Sample tubes/cones of aqueous suspension were shaken vigorously and vortexed on a vortex mixer for 5 min, then ~40 droplets (1 µL volume each) were pipetted onto the hydrophobic glass slide. A Perspex shield was placed over the glass slide to form a chamber, which was purged with dry $N_2$ gas to prevent the formation of condensation on the cold stage. The cold stage was cooled at a rate of 1 °C min⁻¹ to −40 °C and videos were recorded via a webcam as the array of droplets froze, with the droplet freezing temperatures being determined by synchronisation of the

videos with the cold plate temperature measurements. The temperature error of the µL-NIPI was ±0.4 °C (Whale et al., 2015).

**2.4 Processing of droplet freezing assay data**

Droplet freezing assay analysis of the aqueous suspensions of collected ambient aerosol allowed the fraction of the droplets frozen as a function of temperature, $f_{ice}(T)$, to be determined using Equation (1) (Vali, 1971, 1994):


$$f_{ice}(T) = \frac{n_{ice}(T)}{n_{tot}}, \tag{1}$$

where $n_{ice}(T)$ is the number of droplets frozen at temperature $T$, and $n_{tot}$ is the total number of droplets. The fraction frozen values represent the raw data for the droplet freezing assays, and these are shown for all of the campaign

results in Figures S1-S7 in the SI.

Early in the campaign (from 25/10/18 to 30/10/18 (DD/MM/YY)), the bulk of the droplets in the purified water blanks froze in the region of around −23 °C to −29 °C. However, on 31/10/18, a change occurred in the blanks, and from that point until the end of the campaign the majority of droplets froze from around to −21 °C to −26 °C,



thus exhibiting a noticeable increase in the background signal. The cause of this change in the quality of the blanks

could not be identified, despite several possibilities being tested. To compare sample data against the appropriate set

of blanks, the blanks were separated into two categories: (i) "early" blanks analysed between 25/10/18 and 30/10/18,

and (ii) the poorer "later" blanks analysed from 31/10/18 to 04/11/18. The fraction frozen data for these two sets of

blanks are shown in Figure S1 in the SI.

Handling blanks were performed in order to assess the extent of contamination that could occur as part of the

handling of the samples, using the methods described earlier for the filter and impinger samples. The results are

shown in Figure S2 in the SI and show little difference between the handling blanks and the purified water,

demonstrating that there was little, if any, contamination introduced into the samples as part of the handling

procedure.

Controls for the heat test were also performed by heating purified water and testing it before and after the

treatment. Two such tests were performed: one using an "early" blank and one using a "later" blank, and the results

are shown in Figure S3 in the SI. The control test using water from the "early" part of the campaign demonstrated a

small increase in ice-nucleating activity following the heat treatment. However, the control test for the "later" blank

demonstrated that the heated water had a similar activity to the unheated water, and the heated "early" water showed

similar results to both the heated and unheated "later" water. This suggests very clean blanks (e.g. the "early"

untreated water here) are prone to a small amount of contamination upon heating, whereas blanks already containing

some notable contamination are unaffected by further heating. Given this, and the fact that no decrease in activity of

the purified water blanks was observed following heating, the heat test was determined to be viable for the ambient

aerosol samples.

Fraction frozen curves were generated for the aerosol samples, and can be found in the SI: results for the 3h

filter samples are shown in SI Figure S4, the 24 h filter samples are shown in SI Figure S5, the impinger samples are

shown in SI Figure S6, and the heat tests for all of the samples are shown alongside the unheated results in SI Figure

S7. The plots also show the blanks, and the data for both the blanks and the samples are separated into the "early"

and "later" categories so that the samples can be compared against their appropriate blanks.

**2.5 Background subtraction of INP data**

The data shown in the fraction frozen curves were used to perform background subtraction of the blanks from the

sample data in order to remove the influence of impurities from the INP signal. This procedure is described in detail

by Vali (2019) and Sanchez-Marroquin et al. (2021). The first step was to bin the data from all of the individual

droplet freezing assays, both the blanks and samples, into 0.5 °C temperature intervals, and then calculate the

differential freezing nucleus spectrum, $k(T)$ (cm$^{-3}$ °C$^{-1}$), using Equation (2) (Vali, 2019; Vali, 1971):






$$k(T) = -\frac{1}{v \cdot \Delta T} \cdot \ln\left(1 - \frac{\Delta N}{N(T)}\right), \tag{2}$$

where $v$ is the droplet volume (1 μL, i.e. 0.001 cm$^3$), $\triangle T$ is the temperature interval (0.5 °C), $N$ is the number of unfrozen droplets in the temperature interval, and $\triangle N$ is the number of droplets frozen in the temperature interval.

The $k(T)$ values for the background data (i.e. the purified water blanks) were obtained by combining all of the droplet freezing temperatures within the respective "early blanks" and "later blanks" populations. The average and standard deviation of the $k(T)$ values in each temperature bin were determined and these are shown in Figure S8 in the SI.

These average blank $k(T)$ values for the "early" and "later" populations were subtracted from the sample $k(T)$

values for each temperature bin, as shown in Figure S9 in the SI. Uncertainty values were determined by calculating the randomness of active site distribution within the droplet population during the droplet freezing experiments via Monte Carlo simulations (Harrison et al., 2019), the Poisson uncertainties in the data, and the standard deviation of the background $k(T)$ values, with these uncertainties being combined in quadrature to give the final error values for the sample data. Following the background subtraction of the data in terms of $k(T)$, the final values were converted

into the cumulative integrated ice-active site volume density, $K(T)$, a singular approximation, by summing the background-subtracted sample $k(T)$ values for temperatures warmer than $T$, as per Equation (3):

$$K(T) = \sum_{0}^{T} k(T) \cdot \Delta T, \tag{3}$$

The singular approximation assumes that ice nucleation is temperature-dependent and time-independent, that each droplet freezes due to a single nucleation event, droplet freezing occurs at a characteristic temperature depending on the nature of the INPs, and that each droplet contains the same average surface area of ice-nucleating particles. The nucleation of many materials is actually both site- and time-dependent (Knopf et al., 2020; Vali, 2008), but for simplicity we make the assumption here that time-dependence is second order (Vali, 2008; Holden et al., 2019;

Holden et al., 2021) and so apply the singular approximation.

The $K(T)$ values for the samples calculated using Equation (3), for both the "early" and "later" populations, are shown in Figure S10 in the SI. From the background-subtracted $K(T)$ values, the INP concentrations, $N_{\text{INP}}$ (L$^{-1}$ of sampled air), were calculated using Equation (4):

$$N_{\text{INP}} = K(T) \cdot \frac{V_{\text{wash}}}{V_{\text{air}}}, \tag{4}$$

where $V_{\text{wash}}$ is the volume of water used to wash the particles off a collection filter and into suspension or the volume of water in the cone of the impinger (see Table S1), and $V_{\text{air}}$ is the volume of sampled air onto the filter or into the



impinger cone (see Table S1). INP concentrations for both the "early" and "later" samples populations for all samples
are shown in Figure S11 in the SI.

The ice-nucleating activity of the aerosol throughout the campaign was quantified via calculation of the ice-active site density per surface area, $n_s(T)$ (cm$^{-2}$), from $N_{INP}$ using Equation (5):

$$n_s(T) = \frac{N_{INP}}{dS},$$ (5)


where $dS$ is the surface area concentration of particles in the sampled air (cm$^2$ L$^{-1}$), assuming that the total surface area was equal to the surface area of mineral dust. $dS$ was calculated for each filter and impinger sampling time based on the OPC and SMPS data and is provided for each sample in Table S3 in the SI in terms of µm$^2$ cm$^{-3}$.

**2.6 Aerosol monitoring**

Aerosol sizes and concentrations were monitored during the campaign using several instruments. An optical particle counter (OPC; GRIMM Technologies Model 1.109) monitoring in the range of 0.25 – 32 µm was used continuously throughout the campaign and provided data every 6 s. A Scanning Mobility Particle Sizer (SMPS) spectrometer (Model 3938, TSI Inc.) was used to take measurements in the range of 14.1 – 710.5 nm on a daily basis during the filter and impinger sampling periods. The OPC data was used to determine particle concentration (d$N$), and PM$_{10}$,
PM$_{2.5}$ and PM$_1$ concentrations. The OPC and SMPS data were combined to determine the surface area concentrations, d$S$, throughout the campaign.

**2.7 Size-resolved biological analysis**

Size-segregated ambient dust particles were analysed for their bacterial and fungal content as described by Gat et al. (2021). Aerosol particles were collected onto Cyclopore polycarbonate filters every day of the campaign using a
micro-orifice uniform deposit impactor (MOUDI; MSP Corporation model 110-R) (Marple et al., 1991), which operated at 30 L min$^{-1}$ for 8 h, similarly to Huffman et al. (2013) and Mason et al. (2015). DNA was extracted from the filters using the DNeasy® PowerWater® kit (Qiagen) with slight modifications to the manufacturer's protocol: elution was divided into two steps, adding 50 µL of elution buffer (EB) solution and centrifuging at 13,000 g for 1 min each time. The filters were cut in half, and half of each filter was stored as a backup at −20 °C.
Two halves from two sequential MOUDI sampler stages were combined in each PowerWater® bead-tube. Stages 2 ($D_{50}$ = 5.6 µm) and 3 ($D_{50}$ = 3.2 µm) of the MOUDI were combined to give the "coarse" particle size range, stages 4 ($D_{50}$ = 1.8 µm) and 5 ($D_{50}$ = 1.0 µm) were combined to give the "intermediate" particle size range, and stage 6 ($D_{50}$ = 0.6 µm) and 7 ($D_{50}$ = 0.3 µm) refer to the "fine" particle size range. After extraction, the DNA samples were stored at −20 °C for further analysis. Blank filters were extracted similarly.



Total bacterial and fungal concentrations were determined by quantitative polymerase chain reaction (qPCR), using primers listed in Table S4 in the ESI. Bacterial DNA was analysed via amplicon sequencing of the 16S rRNA gene, while fungal activity was analysed using the ribosomal internal transcribed spacer (ITS) region. qPCR reactions were performed as follows: 10 μL of sensiFAST® SYBR mix (Bioline), 1 mM of each primer, 2 μL of extracted DNA, and PCR-grade deionised water (DI) to reach a total reaction volume of 20 μL, with triplicates for each DNA

sample and a negative control (i.e. a blank filter). Amplification was performed in 96-well plates using a StepOnePlus® Real-Time PCR instrument (Thermo Fisher Scientific), as follows: 5 min at 95 °C, followed by 40 cycles of 5 s at 95 °C, and 30 s at 60 °C or 52 °C for 16S or ITS annealing temperatures, respectively. A melting curve stage at 60−95 °C was included, confirming substrate specific amplification. Standard curves were constructed using known concentrations of plasmids containing the relevant genes along with a non-template control.

**2.8 Meteorological data**

Local meteorological data was obtained from the Israel Meteorological Service (IMS, https://ims.gov.il/en) station #42, Bet Dagan: 32.0073 latitude, 34.8138 longitude, 31 m elevation,  approximately 14 km north of the campaign site. This data includes $PM_{10}$ concentrations, temperature, atmospheric pressure, rainfall, wind direction, wind speed, and relative humidity.

**2.9 Air mass back trajectories**

The National Oceanic and Atmospheric Administration (NOAA) Air Resources Laboratory's HYSPLIT (Hybrid Single Particle Lagrangian Integrated Trajectory) model (https://www.arl.noaa.gov/hysplit/) (Stein et al., 2016; Fleming et al., 2012; Draxler and Hess, 1998) was used to generate 72 h back trajectories of the movements of air masses to the sampling region during the field campaign. Back trajectories were obtained for 10:00 local time (8:00

UTC) and 18:00 local time (16:00 UTC) every day of the campaign.

**3 Results and Discussion**

**3.1 Ice-nucleating particle concentrations**

Of the different types of sampling performed (3 h filters, 24 h filters, 20 min impinger samples), the 3 h filter samples provide a more complete and consistent time series for the campaign; hence they are the main results presented here.

The final INP concentrations for the 3 h filter samples, collected every morning and afternoon throughout the campaign, are presented in Figure 1a, and demonstrate a range of variability in INP concentrations during the two-week campaign, from ~0.2 – 14 INP L$^{-1}$ of air in the temperature range of −10 to −25 °C. Concentrations were sufficiently high, and even with the relatively short sampling times and volumes employed here they were largely



above the baseline of the cold stage instrument, enough to be well above the lowest freezing temperature of the instrument (see Figure S9 in the SI), varying between ~0.1 – 2 INP L$^{-1}$ at −20 °C.

Some INP concentrations also exhibited a "hump" of activity at the warmer range of freezing temperatures (in particular the "181029 afternoon" sample), which could be indicative of biological INP activity (Schnell, 1977; Conen et al., 2011; O'Sullivan et al., 2014; O'Sullivan et al., 2018), i.e. a potential "biological hump" in the $N_{INP}$ data. This phenomenon is discussed in further detail later in the manuscript.

The 24 h filter data, shown in Figures S11a and S12 in the SI, shows consistent INP concentrations that reside within the median of the 3 h filter data. This is likely a consequence of averaging the INP data over such a long period of time during changes in $N_{INP}$ between high and low values, for example if there were rapid changes in $N_{INP}$ with diurnal cycles or changes in air masses. This highlights the need for shorter time resolution measurements.

The $N_{INP}$ values from the Coriolis® impinger, measuring particles > 0.5 μm, are shown in Figures S11 and S13 in the SI, were collected on time scale of 20 min (60 min in one case) and showed a large degree of variability depending on the air mass, and this will be described further when discussing the time series of the campaign. Further, as for the 3 h filter results, the "181029 afternoon" exhibited a notable potential "biological hump" on $N_{INP}$ at the warmer temperatures, which will be explored alongside the 3 h filter data later in the manuscript.

Figure 1b illustrates the 3 h filter data shown in Figure 1a but now separated into two populations: "dust events" and "clean days". The categorisation of these populations follows that prescribed by Gat et al. (2021), who performed an analysis of airborne bacterial and fungal communities during the same campaign. Air masses were categorised based on the back trajectories for a given day, as assigned by Gat et al. (2021) previously (see Table S2 in the SI): southwesterly = SW, northwesterly = NW, and easterly = E. Days during which the PM$_{10}$ concentrations were greater than 44 μg m$^{-3}$ were classified as dust events as per Krasnov et al. (2016), and were designated with a lower case "$_D$". Thus, INP measurements taken on days designated as dust events are shown as brown lines in Figure 1b, while those on clean days are shown as blue lines. However, it must be noted that the region is typically dusty regardless, hence the term "clean days" is relative; most "clean days" would still be dusty, but below the 44 μg m$^{-3}$ threshold.

## 3.2 Comparison to literature $N_{INP}$ data

INP concentrations from this campaign are plotted against relevant literature data in Figure 1c. The results demonstrate that, with only a few exceptions at temperatures warmer than ~−19 °C, the dust days had consistently higher INP concentrations than the clean days. The data fits largely well within the boundaries of the global INP data compiled from precipitation sample data from between 1971-2014 by Petters and Wright (2015), with the Eastern Mediterranean data sitting in the middle of the compiled data above ~−24 °C.

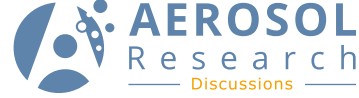

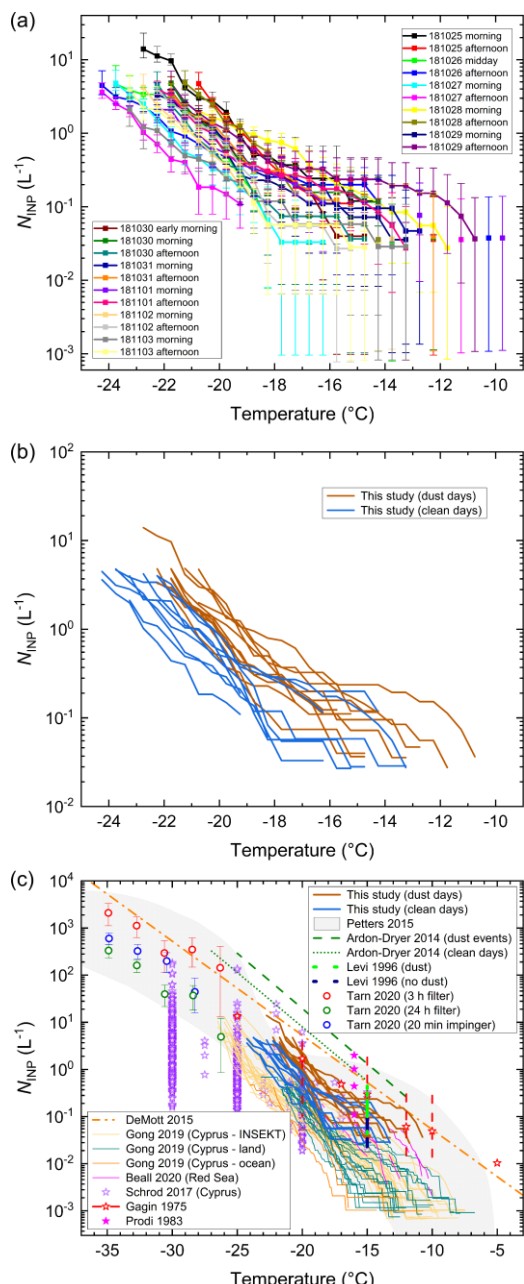

**Figure 1: Plots showing ice-nucleating particle (INP) concentrations ($N_{INP}$) in Israel. (a) Background-subtracted INP concentration spectra for samples collected for 3 h onto filters from 25th October−4th November 2018. The temperature error is estimated to be 0.4 °C (Whale et al., 2015). Dates are given in the YYMMDD format. (b) The same data shown represented as being wither from a dust event (brown) or a clean day (blue). (c) Comparison of INP concentration data from this study to literature data obtained in Israel and in nearby regions of the Eastern Mediterranean. Literature data are represented as discrete data points (Gagin, 1975; Levi and Rosenfeld, 1996; Schrod et al., 2017; Tarn et al., 2020) or as lines (Beall et al., 2022; Gong et al., 2019) as appropriate. The INP concentration range measured using precipitation samples compiled by Petters and Wright (2015) is also shown in grey.**

420





When compared to the INP concentrations measured by Gong et al. (2019) in Cyprus from both land and oceanic
sources at an earlier time in the year (2-30 April 2017), our clean day data is consistent with much of the land sector
data and much higher than the ocean sector data. However, data obtained in the Red Sea (5-7 July 2017), with air
masses flowing over the Nile Delta and the Sinai Peninsula, by Beall et al. (2022) during a ship campaign, compares
well with much of the dust event data from our campaign. Only a portion of the data of Schrod et al. (2017), taken at
altitudes of 717-2,550 m a. m. s. l. over Cyprus using UAVs, is in the same temperature regime as our INP
measurements, but in the ranges of −17.5 °C to −22.5 °C the data sets compare well despite the altitude difference,
although the Schrod et al. (2017) data typically covers a wider INP concentration range.

Aircraft and ground-based measurements by Gagin (1975) in Israel showed INP concentrations similar to
ours at −20 °C, and while there is some comparison at −15 °C and −12 °C, the concentrations tend to trend higher
than all but our most active samples. Levi and Rosenfeld (1996) collected filter samples at ground level for INP
analysis at −15 °C, using a static thermal diffusion chamber, during dust events and clean days. Their results showed
that INP concentrations more than doubled during dust storms.

The data from Tarn et al. (2020) was obtained using three of the same samples collected during the October-
November 2018 campaign: a 3 h filter sample ("181031 afternoon"), a 24 h filter sample ("181025 to 181026 (24
h)"), and an impinger sample ("181103 afternoon (impinger)"). These samples were analysed using the Lab-on-a-
Chip Nucleation by Immersed Particle Instrument (LOC-NIPI). The LOC-NIPI is a continuous flow cold stage
instrument that analyses 100 µm diameter droplets (~520 pL). Although this data does not overlap in temperature
with the µL-NIPI data (using 1 µL droplet arrays), it extends the INP data to colder temperatures and is
complementary to the 3h filter results shown here.

The "dust event" and "clean day" data of Ardon-Dryer and Levin (2014) from ground-based measurements
is much higher in concentration compared to all other available literature data from the region, but showed similar
trends to the those illustrated here. The INP parameterisation of DeMott et al. (2015) for mineral dust particles larger
than 0.5 µm was calculated based on the OPC data and is shown in Figure 1b, but is much higher than the bulk of the
INP data from the region.

**3.3 Time series of the Eastern Mediterranean campaign**

A time series of multiple data sets for the duration of the campaign in Rehovot, Israel, from 25[th] October to 3[rd]
November 2018, is provided in Figure 2. Meteorological data obtained from the IMS station in Bet Dagan (11 km
away) is shown in Figures 2a-e. Aerosol data measured using the GRIMM OPC is provided for particle concentrations
($dN$) (Figure 2f), $PM_{10/2.5/1}$ mass concentrations (Figure 2g), and particle size distributions in terms of $D_p$ *vs.*
$dN/dlogD_p$ (Figure 2h). Raw data from the GRIMM OPC and the SMPS are provided in Figures S14 and S15 in the
SI, respectively. A higher resolution version of the $D_p$ *vs.* $dN/dlogD_p$ plot is shown in Figure S16a in the SI, while the
surface area distribution ($dS$) in terms of $D_p$ *vs.* $dS/dlogD_p$ is also shown in Figure S16b in the SI.



**Figure 2: Time series of meteorological and aerosol data for the field campaign in Rehovot, Israel, with dates in the DD/MM/YYYY format. The air mass category for each day is provided at the top (e.g. SW_D, NW1…). 3 h filter sampling times are shown as vertical purple bars. (a-e) Meteorological data from the Israel Meteorological Service (IMS) station at Bet Dagan: (a) Relative humidity. (b) Atmospheric pressure. (c) Temperature, (d) Rainfall, (e) Wind speed, with wind direction represented by coloured arrows: westerly winds are shown in shades of red and easterly winds in shades of blue. (f) Total aerosol particle concentration, d$N$, determined using the OPC. (g) PM$_{10}$, PM$_{2.5}$ and PM$_1$ mass concentrations from the OPC data, alongside PM$_{10}$ concentrations from the IMS station. (h) Particle size ($D_p$) distributions (d$N$/dlog$D_p$). (i) Background-subtracted ice-nucleating particle concentrations ($N_{INP}$) at −14 °C, −16 °C, −18 °C, −20 °C, and −22 °C.**




INP concentrations at selected temperatures were calculated from the 3 h filter data in Figure 1a and are illustrated in Figure 2g, while the sampling times are shown as purple bars throughout the figure. The air mass category and ID developed by Gat et al. (2021) for the campaign is provided for each day at the top of the image based on wind directions and air mass back trajectories. HYSPLIT air mass back trajectories for 10:00 and 18:00

local time are provided in Figure S17 in the SI, and are in good agreement with those of Gat et al. (2021), who used a Lagrangian analysis tool, LAGRANTO 2.0 (Sprenger and Wernli, 2015) for analysis of the same time periods.

The campaign began with the tail-end of a Saharan dust storm event from the southwest (SW$_D$) (Figure 2e) on the 25$^{th}$ October, with the air mass also passing over the fertile land of the Nile Delta (Figure S17 in the SI). This air mass yielded the highest aerosol concentrations throughout the campaign, with particle surface area concentration

modes for particles of ~0.2 µm and ~4 µm diameter, and some of the highest INP concentrations throughout the campaign. This was immediately followed by an intense rainstorm later that same night (Figure 2d), which saw aerosol concentrations plummet for the following two days with the air masses coming from the northwest through Central and Eastern Europe and over the Mediterranean Sea. INP concentrations at the colder temperatures (−20 °C and −22 °C) dropped significantly on the 26$^{th}$ October compared to the SW$_D$ event, while they remained somewhat

comparable at warmer temperatures (−16 °C and −18 °C). Concentrations at all of these temperatures dropped further on the 27$^{th}$ of October during the same NW trajectory period, although there were multiple changes in wind direction throughout those sampling times as shown in Figure 2e which may have influenced those results. The 20-60 min impinger samples from the 26$^{th}$ and 27$^{th}$ of October exhibited the lowest INP concentrations in the impinger data (Figure S13 in the SI), consistent with the data from the 3 h filter samples.

The 28$^{th}$ to the 31$^{st}$ of October saw the air masses coming from the east and carrying dust from Syria (the E$_D$ category). This saw an increase in particle concentrations and surface area concentrations for larger particle sizes, in addition to higher INP concentrations, similar to those in the SW$_D$ event. INP concentrations (at −18 °C to −22 °C) remained consistently high throughout this 4-day E$_D$ period. Interestingly, while the 72 h air masses back trajectories on the 30$^{th}$ and 31$^{st}$ October came almost entirely over land from the east, those on the 28$^{th}$ largely came from Europe,

passing over a portion of the Mediterranean Sea, and only spent a relatively short time over the land to the East of the sampling location. Impinger samples were collected for 20 min during the afternoons of the 28$^{th}$ to 30$^{th}$ October and these showed high INP concentrations (Figure S13 in the SI), again consistent with the 3 h filter samples.

The afternoon of the 29$^{th}$ October was of particular interest, with notably high INP concentrations at −14 °C to −16 °C, with the 20 min impinger sample from that time yielding the highest INP concentrations of the impinger

dataset. Here, the air mass came from the North, passing over the eastern side of the Black Sea and over the region known as the Fertile Crescent, which spans the northern regions of Israel, Syria, and Iraq. The Fertile Crescent is the historical origin of agriculture and animal herding that started around 12,000 years ago (Salamini et al., 2002), though in modern times has suffered from long-term drying and is increasingly prone to drought (Kelley et al., 2015; Zittis



et al., 2022), but still remains an active location for agriculture and vegetation (Zaitchik et al., 2007). Given this
unique air mass trajectory amongst the rest of the dataset, the high INP concentrations at warmer temperatures, and
the "hump" in the INP spectrum that may indicate biological INP content as mentioned earlier, this data is hence
discussed in further detail later in the manuscript.

Following the $E_D$ dust event, the remainder of the campaign from 1st to 4th November was characterised by
easterly winds and air mass back trajectories that were similar to the preceding two days, but with lower aerosol
particle concentrations. With this decrease in aerosol concentration came a steady decrease in INP concentrations at
the colder temperatures, and a general trend of INP concentrations in the afternoons being higher than those in the
morning. The 20 min impinger samples from the 2nd and 3rd November also showed decreasing activity following the
$E_D$ event (see Figure S13 in the SI).

### 3.4 Potential biological INP content

As discussed earlier, some INP concentrations in Figure 1a showed a "hump" of higher activity in the spectra at
warmer temperatures, which may be indicative of biological INP activity that could be associated with dust or soil
particles (O'Sullivan et al., 2014; Conen et al., 2011; Schnell, 1977; Schnell and Vali, 1976).The most notable sample
spectra exhibiting this potential "biological hump" was from the afternoon of 29th October (Figure 1a). That specific
air mass had passed over the Fertile Crescent to the north of the sampling site during the easterly dust event ($E_D$2).
This was corroborated by a similar hump in the 20 min impinger data during that same timeframe (see Figure S18 in
the SI).

By coincidence, the sampling on the afternoon of the 29th October was performed simultaneously with two
BGI PQ100 filter samplers: one set up for $PM_{10}$ and one set up for $PM_1$. The results for these can be seen in Figure
1a, with a more direct comparison in Figure S18 in the SI alongside the impinger data from the same timeframe. The
$N_{INP}$ of the $PM_1$ aerosols was relatively high, though also notably lower than for $PM_{10}$, suggesting that while much
of the ice-nucleating activity was due to particles in the 1-10 µm size range, there was still a lot of activity caused by
particles <1 µm. The $PM_1$ data also showed the same "biological" hump as in the $PM_{10}$ data.





**Figure 3: Plot showing the effect of heat treatment (95 °C for 30 min, as per Daily et al. (2022)) on the ice-nucleating activity**
**of aqueous particle suspensions obtained from 3 h filter samples. This heat treatment is used as an indicator for the**
**potential presence of proteinaceous INPs should there be a notable loss of activity at warmer temperatures following the**
**heat treatment. Dates are given in the YYMMDD format.**





As a further means to test for biological ice-nucleating activity, we performed heat treatments (95 °C for 30 min) (Christner et al., 2008a; Daily et al., 2022) on nearly every sample collected and compared the ice-nucleating activities
of the original samples to the heated samples in Figure 3 for the 3 h filter samples. The heat test results for the 24 h filter samples and the impinger samples are provided in Figure S19 and Figure S20 in the SI, respectively.

Heat treatment of the sample from the afternoon of 29th October demonstrated a significant loss in activity at the warmer parts of the $PM_{10}$ and $PM_1$ curves (Figure 3), i.e. a loss of the "hump" in each case (also see Figure S18 in the SI). The heat treatment of an impinger sample also showed a flattening of the hump (see Figure S18 in the SI),
though not a similar dramatic loss in activity as for the filter samples. The heat treatment is not a definitive test for biological material, but since we expect the mineral ice-nucleating activity to be controlled by K-feldspar in this region (Boose et al., 2016) we believe it to be a reasonable indicator that the "humps" in the $N_{INP}$ data were due to biological INP activity given the caveats of the procedure described by Daily et al. (2022). Thus, it is possible that biological INPs may have been entrained into the air mass as it passed over the Fertile Crescent to the north of the
sampling site, suggesting that the Fertile Crescent could be a potential source of high temperature, heat-labile INPs for the Eastern Mediterranean and Middle Eastern regions.

While not as pronounced, the $N_{INP}$ data from the morning of the 29th October also exhibited a notable hump that was lost following heat treatment, with the air mass for that sample having passed over a small region of the Fertile Crescent to the north of the sampling site. No other sampled air masses (for 500 m AGL) passed over the same
area throughout the campaign, making these measurements unique, but the hypotheses therefore somewhat speculative. However, Gong et al. (2019) noted a potential biological signal in their INP concentrations at temperatures warmer than −15 °C during their campaign in Cyprus and suggested it originated locally, but we note that a handful of their air mass back trajectories appeared to pass over the Fertile Crescent and the Nile Delta regions.

A handful of other $N_{INP}$ spectra potentially displayed smaller potential biological humps and were found to
be sensitive to heat treatment (Figure 3). The sample for the afternoon of 25th October also comprised from a unique air mass for the campaign, originating from the west of Italy before travelling over north of the Sahara Desert and passing over the Nile Delta, the southeasternmost part of the Fertile Crescent, which may have contributed biological INPs to the sample. This would also agree with the findings of Beall et al. (2022) during the Red Sea leg of their cruise when they encountered air masses from the Nile Delta that showed heat-labile, likely biological, INP content.
The sample from the afternoon of 26th October also demonstrated a small degree of heat sensitivity following the rain event that washed a lot of dust from the air, with the following air mass passing over Central/Eastern Europe and the Mediterranean Sea prior to reaching the sampling site.

Samples for the 1st (morning) and 3rd (afternoon) of November, obtained during the easterly air mass event (E1-E3) may have shown a minor decrease in $N_{INP}$ following heat treatment in the 3 h filter and/or the impinger
samples, and the air masses did not appreciably differ from those throughout the rest of the easterly event. We do note, however, that the Fertile Crescent extends to eastern Iraq and western Iran, and so could potentially have had

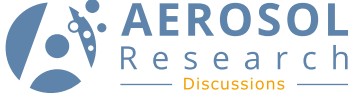

an influence on air masses encountered by the sampling site, although the contribution of biological INPs here was relatively minor, particularly compared to the results from the air masses that passed over the northern Fertile Crescent ($E_D2$).


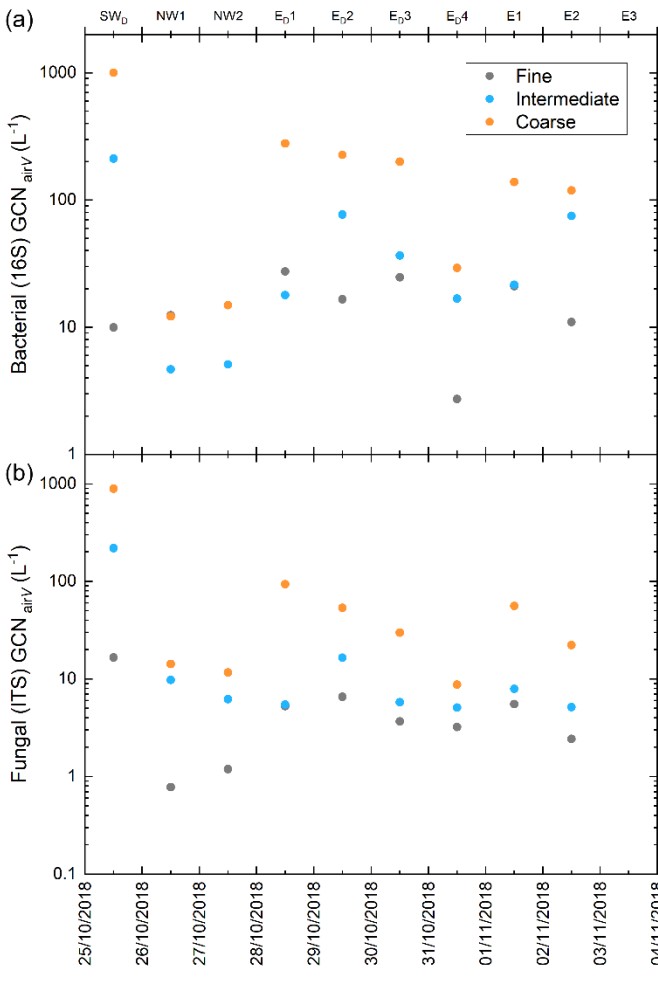

**Figure 4: Plots showing size-resolved biological aerosol particle concentrations, in terms of gene copy number (GCN) per litre of air, for (a) bacteria, analysed using 16S ribosomal RNA, and (b) fungi, using the ribosomal internal transcribed spacer (ITS) region. Particle size ranges were stratified into: coarse, intermediate and fine.**


Israel has varying airborne microbiomes that depend on air mass, and Gat et al. (Gat et al., 2017; Gat et al., 2021; Erkorkmaz et al., 2023; Peng et al., 2021) have measured a variety of desert- and soil-associated bacteria and fungi in air masses that passed over the Nile Delta and the Arabian Peninsula, with some of the latter cases also





containing agricultural soil bacteria. Samples from days with no dust events mostly contained bacteria associated with human activity. Interestingly, dust storms were found to lower the proportion of fungi in the airborne microbiome of the region (Peng et al., 2021).

Throughout the October-November 2018 measurement campaign, Gat et al. (2021) performed an in-depth analysis of the microbiome of the air masses via amplicon sequencing of the 16S rRNA gene and ribosomal internal
transcribed spacer (ITS) region to study size-resolved bacteria and fungi, respectively. Their results showed that bioaerosol communities varied with air mass, and that there was significant transport of bacteria as aggregates or attached to dust particles during the SW$_D$ and E$_D$ dust events (Erkorkmaz et al., 2023; Gat et al., 2021) . Bacterial communities, particularly those of single cells smaller than 0.6 µm, were comprised of mixtures from local and transported sources. Further analysis of that data, presented here, demonstrates the highest concentration of airborne
fungal (ITS) gene copy numbers (GCNs) across the three particle size ranges (fine, intermediate, coarse) during the SW$_D$ event on the 25$^{th}$ of October, where the air mass had travelled over the Nile Delta, while the bacterial (16S) concentrations were the highest in both the coarse and intermediate modes during that event.

The GCN data for bacteria and fungi during the 29$^{th}$ of October appear unspectacular for the coarse mode of particles, but closer inspection reveals that the fine and intermediate fungal concentrations were higher than all other
days except for the 25$^{th}$ of October. The bacterial concentrations in the intermediate size range were the second highest of the campaign, while the fine particle concentrations relatively high. The biological content from the Easterly direction air masses (1$^{st}$ and 2$^{nd}$ October) was also relatively high, particularly in the coarse mode.

While these values are for total populations rather than populations known to have ice-nucleating activity, together with the heat treatment data and the findings of Gat et al. (2021), this data further supports the hypothesis
that biological INPs may have been entrained in air masses that passed over the Fertile Crescent, particularly the Nile Delta region and the northern Syria region.

### 3.5 Ice-active site surface density, $n_s(T)$

The ice-nucleating activity of the aerosol in the region throughout the campaign was quantified via calculation of the ice-active site density per surface area, $n_s(T)$ (cm$^{-2}$), from the INP concentrations (L$^{-1}$) using Equation (5). The
resultant spectra for $n_s(T)$ are shown in Figure 5a, while Figure 5b shows the same data separated into "dust events" and "clean days". Figure 5c provides a comparison of these $n_s(T)$ values to the relevant literature, here with the campaign data separated into "dust events" (SW$_D$ and E$_D$ air masses) and "clean days" (NW and E air masses). Most of the data for the campaign lie on top of each other at the colder temperatures ($< -18$ °C), with some of the samples from the E air mass (clean days) being lower than the bulk of the data.






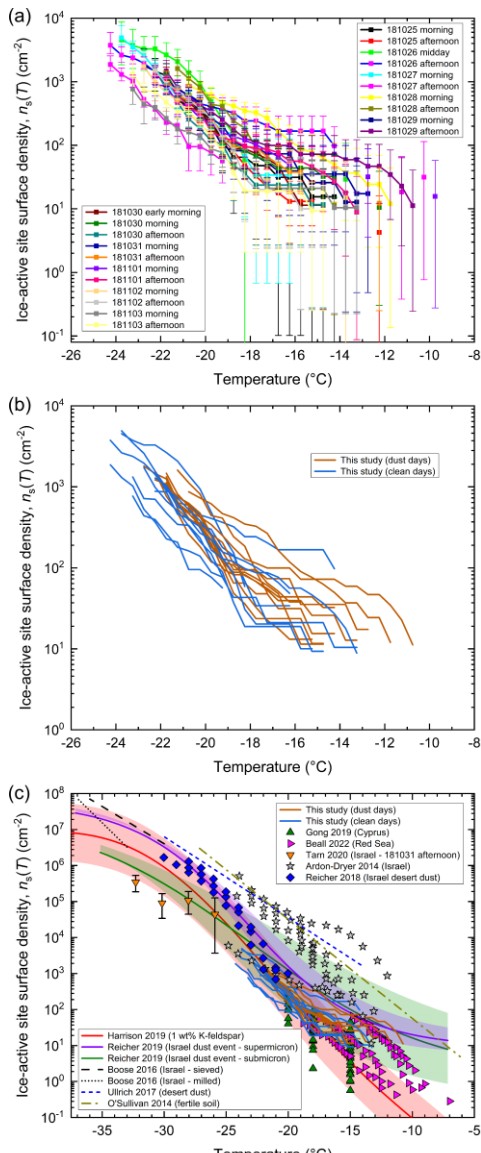

**Figure 5: Plots showing the ice-active site surface density, $n_s(T)$ for INPs in Israel. (a) Background-subtracted $n_s(T)$ spectra for 3 h filter samples. Data points consistent with the background are shown as hollow data points and are considered to be upper limits. Dates are given in the YYMMDD format. (b) The 3 h filter sample data separated into dust events (brown) and clean days (blue). (c) Comparison of $n_s(T)$ values to relevant literature, including $n_s(T)$ data from Israel and the East Mediterranean region (Ardon-Dryer and Levin, 2014; Beall et al., 2022; Gong et al., 2019; Reicher et al., 2019; Reicher et al., 2018; Tarn et al., 2020). Several parameterisations for mineral dust are provided, including those for surface samples from arid locations around the world (Ullrich et al., 2017), sieved and milled surface samples from Israel (Boose et al., 2016), supermicron and submicron particles in desert dust storms (Reicher et al., 2019), the Harrison parameterisation for K-feldspar assuming K-feldspar comprised around 1 wt% of airborne mineral dust (Harrison et al., 2019), and O'Sullivan parameterisation for fertile soils (O'Sullivan et al., 2014).**



Our data was around 2 orders of magnitude smaller than the parameterisation of Ullrich et al. (2017) derived from soil samples and precipitated dust samples collected from a number of arid locations, including Israel and the Nile

Delta. However, the bulk of the data below −18 °C match very well with the K-feldspar parameterisation developed by Harrison et al. (2019), here being scaled to 1 wt% (noting that Boose et al. (2016), also plotted here, found 1.7 ± 0.4 % wt% and 1.3 ± 0.4 % wt% K-feldspar in sieved and milled surface dust from Israel, respectively). The data at temperatures warmer than about −20 °C lies increasingly higher than Harrison et al. (2019) and within the lower uncertainties of the two parameterisations of Reicher et al. (2019) for supermicron and submicron dust during dust

events. Some samples overlap even more with the Reicher et al. (2019) regimes, such as "181029 afternoon" which likely had more influence from biological INPs. Sample "181026 afternoon" demonstrated a high $n_s(T)$ value since particle concentrations had dropped following a rain event, and thereafter may have been influenced by bioaerosols in the air mass passing over Europe and the Mediterranean Sea.

       Our data overlapped with some of the higher $n_s(T)$ data of Gong et al. (2019) from Cyprus, which followed

the Harrison et al. (2019) K-feldspar into the warmer temperatures. However, the data of Beall et al. (2022) from the Red Sea matched ours very well. As for the $N_{INP}$ comparison, the Israel data of Ardon-Dryer and Levin (2014) was largely an order of magnitude or so higher in $n_s(T)$ than our observations, with only one of their clean day datasets being comparable though their data does compare well with the Ullrich et al. (2017) parameterisation for desert dust. The fertile soil parameterisation from O'Sullivan et al. (2014) is plotted and also sits much higher than our data, being

somewhat more similar to the Ullrich et al. (2017) parameterisation.

       Some of the collected data compared well at the colder temperatures with that of Reicher et al. (2018) for dust storms. The data shown from Tarn et al. (2020) represents a 3 h filter ("181031 afternoon"), as analysed using the LOC-NIPI microfluidic platform, and is complementary to the datasets obtained using the μL-NIPI for this campaign.

That the $n_s(T)$ values for the clean days and the dust event days largely collapsed on top of each other, particularly at temperatures < −18 °C, is not surprising given the atmosphere always contains dust particles and so the term "clean" is relative only to the high loadings measured during elevated dust events. Likewise, the activities below −18 °C fall onto the Harrison et al. (2019) parameterisation for 1 wt% K-feldspar is consistent with the INP population being dominated by K-feldspar mineral dust, with samples exhibiting increased $n_s(T)$ at warmer

temperatures potentially being due to biological content.

       However, the "cleanest" day was the 26[th] October following an intense rain event the previous night (see Figure 2d) that resulted in low aerosol concentrations, yet surprisingly high INP concentrations considering. Interestingly, the midday INP concentrations from a more westerly wind direction were notably higher than those from the afternoon more northwesterly direction (see Figure 2e). The surprisingly high INP concentrations following



the rain event resulted in high $n_s(T)$ activities given the low total aerosol concentrations, the reasons for which are unclear.

Overall, these results bring us to a similar conclusion as Gong et al. (2019) and Beall et al. (2022),  that, while $n_s(T)$ parameterisations for mineral dust may be relevant for field samples below around −15 to −18 °C when the population is expected to be dominated by K-feldspar, they are not representative of regional INP activity at
temperatures much warmer than −18 °C when the INP population contains substantial biological content. Therefore, as suggested by Gong et al. (2019) and Beall et al. (2022), it may be that other methods of representation of INP activity are warranted. This could include the use of probability density functions (PDFs) from observed INP concentrations, as suggested by Gong et al. (2019), or $n_s(T)$ parameterisations for specific regions and events/seasons, such as those of Reicher et al. (2019) for dust events in Israel that here captured both the K-feldspar INP population
but also the potential biological INP population indicated in these results.

## 4 Conclusions

The Eastern Mediterranean is an interesting location for aerosol and INP analysis, encountering air masses from Europe, sea-spray aerosol from the Mediterranean Sea, and dust storms from the Sahara and Arabian Deserts, with the local atmosphere always containing some degree of dust loading. Given such variation in sources and transport
of INPs, it is important to try to understand their potential and comparative effects on the region. Here, we undertook an intensive two-week INP sampling campaign in Israel to monitor concentrations and activity every morning and afternoon, with coincident measurements of local aerosol concentrations. Four main air masses were encountered during the campaign, each offering some insights into the INP populations in the region. A Saharan dust event from the south-west yielded high INP concentrations that notably featured a potential biological component, possibly as a
result of passing over the fertile Nile Delta shortly prior to reaching the sampling site. Northwesterly winds that brought an air mass from central/eastern Europe and crossed a section of the Mediterranean Sea brought very low concentrations of aerosols and INPs. Easterly winds brought air masses from Syria and the Arabian Desert with both high (dust events) and low aerosol loadings. The high loadings yielded high INP concentrations that tailed off as the dust loading of the air masses decreased over several days. Some of these samples from the Arabian Desert region
may have had a slight biological INP component in some cases but this was inconclusive.

The final and most remarkable air mass travelled from the north before turning west towards the sampling site, notably passing over the Fertile Crescent. The Fertile Crescent has undergone substantial changes in the last century that have resulted in considerable drying and droughts, particularly since the 1950s when irrigation of the marsh region diverted waters from the Tigris and Euphrates rivers ("Fertile Crescent" (History, 2017; National-
Geographic, 2022)). This was further exacerbated in Iraq in 1991 when Saddam Hussein's government built dams and dikes to drain the marshes, although many dams were later broken following the regime change in 2003 that





allowed waters to flood back into parts of the region (Reuters, 2016). While the region is far less fertile than it once was, it still comprises a vast region of lands and dried soils that contain bacteria and fungi, in addition to farmlands, wetlands, rivers and marshes. It is possible that this change in land use can cause the soils to be more mobile and

sensitive to winds. For example, the presence of dams, water transfer policies and extensive droughts throughout the region have been linked to an increase in dust aerosol optical depth in the western and southern regions of Iran (Ghasem et al., 2012; Hamzeh et al., 2021). Iraq saw an increased frequency of dust events from 1980-1993, a decreased frequency from 1993-2001, and an increase again after 2001 (Attiya and Jones, 2020).

The INP concentrations measured during the two-week campaign in the Eastern Mediterranean in 2018 were

not only high, but heating analysis pointed towards a large biological component of INPs active at warmer temperatures. Given the broad territory that the Fertile Crescent covers, including Israel, Lebanon, Syria, and Iraq in the east, it is possible that some small amount of biological ice-nucleating activity encountered in some of the Easterly air masses also originated from these fertile lands.

In all, INP activity in the region throughout the campaign was largely consistent with being controlled by ~1

% K-feldspar mineral dust at temperatures below about −15 to −18 °C. However, the measurements associated with each of the four air masses suggested that the INP population in Israel can be influenced by biological INPs activated at warmer temperatures, whose origins are associated with fertile lands around the Eastern Mediterranean and the Middle East, most notably the Fertile Crescent and including the Nile Delta. To our knowledge, this is the first time that the Fertile Crescent, apart from a discussion of the Nile Delta by Beall et al. (2022), has been discussed as a

potential source of high-temperature biological INPs for the region around it, likely due to a lack of measurements associated with that particular location.

While this hypothesis is still speculative given the small number of relevant sampling events, the capacity for these fertile lands periodically influencing INP populations in the surrounding areas is compelling, particularly should the current long-term drying trends continue and so alter the INP populations. Thus, we believe that the Fertile

Crescent regions warrant more dedicated analyses of their associated INP populations in order to understand their potential influence on the vast surrounding regions.

Our results also bring us to similar conclusions to those of Gong et al. (2019) and Beall et al. (2022), in that $n_s(T)$ parameterisations for mineral dust are not necessarily appropriate for estimating INP activity in models when INP populations active at warmer temperatures may dominate the mineral population and skew $n_s(T)$ values based on particle surface area.

Other methods of INP representation may be required, or a database of regional and seasonal $n_s(T)$ or $N_{INP}$ parameterisations based on observations.

**Conflicts of interest**

One of the authors is a member of the Editorial Board of *Aerosol Research*.



**Acknowledgements**

The authors thank the Weizmann–UK Making Connections Program, the European Research Council (ERC; grant no. 648661 MarineIce) and the Engineering and Physical Sciences Research Council (EPSRC; grant no. EP/R513258/1) for funding. Grace C. E. Porter is thanked for assistance with the INP background subtraction calculations. Burak Adnan Erkorkmaz is thanked for discussions.

**Data availability**

The data sets for this paper will be made publicly available in the University of Leeds Data Repository (https://archive.researchdata.leeds.ac.uk/).

**Author contributions**

MDT, NR, YR and BJM designed the study. MDT, YR and BJM obtained funding for the study. MDT, NR, MA, SNFS and ADH organised the logistics of the campaign, and prepared and tested equipment for the study. MDT, BVW, MA, NR and DG
performed the experiments. MDT, BVW, MA, NR, DG and AS-M analysed the data from the study. YR and BJM supervised the study. MDT, NR, YR and BJM wrote the paper, and all authors contributed to editing.

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
