# Peer review of "Atmospheric ice-nucleating particles in the Eastern Mediterranean and the contribution of mineral and biological aerosol"

_Aerosol Research, 2023_

## Author Response (AR1)

We thank the reviewers for their comments and critical assessment that have helped to improve the paper. We have addressed all comments and made changes to the manuscript and supplementary information as advised.

**First reviewer's report (RC1)**

*1. The authors suggest that the highest IN concentration (29th of October afternoon) at high temperatures (-14°C, figure 2) might be originating from the "fertile crescent" region. A region rich in agriculture suffering from droughts makes this region a possible emitter of biological IN. However, I cannot fully support this hypothesis based on the backward trajectories presented in the supplement. Multiple of the presented backward trajectories seem similar, for example, the air masses from the 28th of October or 29th of October (before noon), with the latter showing the lowest IN concentration at -14°C. Still, suppose this proves to be a unique trajectory. This singular data point building the whole paper story seems to be a shortcoming given this comprehensive dataset, but to the authors' credit, they mention this in the conclusion. I strongly recommend changing the title of the manuscript or arguing in more detail why this should be the main point of this paper and moving the highlighted backward trajectories into the manuscript.*

Response:

We agree that building the paper around this single data point would be a shortcoming, and because of this we have tried to make clear that this hypothesis is speculative and that, rather than stating that the region is definitely important. Instead we suggest that it warrants a more dedicated investigation in the future. As such, combined with the results from the air mass passing over the Nile Delta, also noted in Beall et al. (2022), we still believe it is worth discussing the role of fertile soils dusts as an intriguing potential influence on the atmospheric INP population. However, we appreciate that the emphasis could be scaled back somewhat and so have reworded various parts to clarify this. We have moved the air mass back trajectory from the 29th October to the main paper, and have changed the title to remove mention of fertile soils. Instead of mentioning fertile soil dusts we have highlighted the biological contribution to the INP population, which is unambiguous: "*Atmospheric ice-nucleating particles in the Eastern Mediterranean and the contribution of mineral and biological aerosol*".

Regarding the comparisons of the back trajectories, we note that only the air masses at much higher altitudes (shown in blue and green) passed over the Fertile Crescent region in the north, while the on at the lower altitudes came from Europe and passed over the Mediterranean, before circling the desert region prior to sampling. Therefore, while there was some interaction with the region of interest, it was certainly not as strong or clear as that in the afternoon. Further, while the INP concentrations at -14 °C for the morning of the 29th October were somewhat low, only around half of the samples collected during the campaign showed any activity at that temperature at all, and of those that did the concentration for that sample was similar to all others apart from the one on the afternoon of the 29th and, to a lesser extent, the one from the morning of the 28th. We also note that the morning of the 29th showed a large decrease in INP activity following heat treatment, and this is discussed in the paper.

*2. All figures in the pdf have a low resolution. I strongly recommend enhancing the quality of the final version.*

Response:

The original figures are high-quality TIFF images but the conversion process to PDF has unfortunately drastically reduced the resolution. Higher quality images have been used in the revised documents and checked following the PDF conversion, while the original high quality figures will be provided separately to the journal.

*3. Section 2.4: The background shift is unpleasant but is usual for field campaigns. As additional information, the authors might describe how the water was stored since this has a curial impact.*

Response:

We have added this information about this in the first paragraph of Section 2.1.

4. *Section 2.7: Why were the samples not measured individually? Combining the data could also have been done afterward.*

Response:

The samples were combined to ensure that there was enough DNA to amplify and detect; if only one sample was used then there may not have been enough DNA to analyse. This has now been explained in the main text.

5. *Figure 1c is quite crowded, making it difficult to extract information.*

Response:

We have removed one of the Gong et al. 2019 datasets (INSEKT) since the remaining data largely covers the same region and shows the differences between the land and ocean sectors. We have reduced the thickness of some of the lines and data points in the literature data, whilst making the lines for our own data thicker to better demonstrate where it sits. We have also remade Figure 1b so that it has the same scale as Figure 1c, and now includes the Petters & Wright (2015) envelope of data, in order to make it easier to see where our data fits amongst the other data in 1c.

6. *Figure 2:*

   *e) the arrows on the 26th of October are not distinguishable from the bars marking the 3-hour filters, making it hard to distinguish the wind direction.*

   *i) Error bars would improve this plot.*

Response:

Figure 2e: The arrow colours are different from the original TIFFs, an unfortunate consequence of the PDF conversion. Higher quality images are now being used, and we have changed the colours to brown/orange/green to reduce the risk of overlap with the 3-hour filter bars.

Figure 2i: We have now included error bars in the plot.

7. *Figure 4: The plot could represent the data better. It took me quite some time to understand this plot. Connecting the points with dotted lines would make it easier to follow the time series. Error bars would also be beneficial.*

Response:

We have connected the points as suggested, and have included error bars.

8. *Line 130: Missing a word: "…, proposed that mineral dust parameterizations alone may not be suitable …".*

Response:

Corrected.

9. *Line 188: One of the closing brackets is redundant.*

Response:

The redundant bracket has been removed.

10. *Line 367 and 451: Is the weather station 14 or 11km away?*

Response:

Line 451 has been corrected to 11 km.

11. *Line 467: It should be Figure 2i instead of 2g.*

Response:

Corrected to 2i.

**12. Line 592: 1st and 2nd November instead of October.**

Response:

Corrected to November.

**Second reviewer's report (RC2)**

*1. Introduction: While the introduction provides an extensive review of the literature on INP measurements in the Eastern Mediterranean, the organization of the introduction could be greatly improved. Currently, much of the introduction is structured as a series of short paragraphs, each summarizing one article, which makes this section choppy and very dry. Is it possible to organize the introduction in such a way that the major findings that have emerged from the entire body of available literature are highlighted, rather than focusing on each article individually? This would make the introduction much more useful to the reader.*

Response:

The latter half of the Introduction has been rewritten in such a way as to make the major findings of previous studies more succinctly presented. It still somewhat follows an article-by-article focused but the literature has now been grouped in a manner to highlight the key findings surrounding specific aspects of the INPs in the region and how they are related between research studies.

*2. Determination of the particle surface area: There needs to be more detail provided regarding how the particle surface area is calculated, as the underlying calculation can have an important impact on the results. For instance, the OPC and SMPS have overlapping ranges, so which instrument was used for those sizes? Was there any attempt to convert from optical and electrical mobility diameters to physical diameters? Mineral dust is not likely to be spherical, but I assume the calculation of surface area assumes sphericity. How is this potential source of error accounted for?*

Response:

We have rewritten large parts of sections 2.6 and 2.7 to provide this information. We now provide the ranges of the OPC and SMPS data when using the two combined, and provide details on the conversion of the SMPS data from mobility to volume equivalent diameters. We have further recalculated the surface area concentrations and their uncertainties based on a dynamic shape factor of 1.1 +/- 0.1, since 1.1 offered the greatest agreement between the OPC and SMPS data in terms of dN/dlogDp and dS/dlogDp, while on occasion values of 1.0 and 1.2 offered good matches and the latter accounts better for mineral dusts. We have also calculated PM10 mass concentrations based on a particle density of 2.65 g/cm3 and plot these in Figure 2g alongside the previous values determined for a density of 1 g/cm3; the value of 1 g/cm3 provides a better match with the data provided by the Israeli Ministry of Environment (IME) PM10 monitoring station, while the value of 2.65 g/cm3 is more representative of mineral dust. The PM10 measurements from the IME website are obtained using a high volume sampler at the monitoring site, presumably via gravimetric filter analysis although we are unable to confirm this.

*3. Air mass back-trajectories: Much more detail is needed regarding these calculations especially because they are fundamental to the authors' conclusions. Why was 72 hours selected for the duration? What meteorological data were used? At what resolution? What altitudes were used for initiating the back-trajectories? Do the classifications of the air mass use only the lowest altitude trajectory or all the trajectories for a given time/date?*

Response:

The 72-h duration of the backward trajectory is a consensus and was chosen in order to asses long range transport of the air masses while limiting the trajectory error (Kahl and Samson, 1986, Karaca et al. 2009, Sari et al. 2020). The trajectories were calculated for 3 arrival heights (500m, 1500 m, and 2500 m), because mineral dust is transported below 3 km in this season (Gobbi et al., 2004). In most cases all three trajectories of the different arrival heights travelled in the same path and therefore classification was based on the three of them. However, even if only one trajectory originated in dust source area, and high PM10 were monitored, it will be defined as the dust origin. We have now added statements describing this to the main text with appropriate citation.

**4.  Page 1, Line 26: The word "representation" is used twice in the same sentence.  Consider rephrasing.**

Response:

The second instance of this has been changed to *"...a deficiency in our understanding of..."*.

**5.  Page 2, Line 40: Consider replacing "INP communities" with "INP populations".**

Response:

We have changed the phrase as suggested.

**6.  Page 4, Lines 108-110: Consider rephrasing this sentence. The sentence structure is not correct.**

Response:

The sentence has been rewritten entirely as part of the changes to the Introduction.

**7.  Page 4, Lines 125-126: Verify sentence.  It seems that there is one or several words missing.**

Response:

This section has been reworded and the sentence removed.

**8.  Page 6, line 188: Please provide reference for size collection efficiency of the Coriolis Sampler**

Response:

We have now included a reference to the Coriolis brochure/specifications, and reference to Carvalho et al. (2008) who tested the Coriolis impinger.

**9.  Page 12, Line 358: Please provide volume used for each primer, not only concentration.**

Response:

We have now included the volumes for each primer and also for the deionised water.

**10. Page 13, Line 388: Consider deleting the phrase "i.e. a potential "biological hump" in the NINP data".  It doesn't seem necessary.**

Response:

We have removed this phrase.

**11. Page 19, Figure 13: Please indicate in caption what ranges the box and whisker plots indicate.**

Response:

We have now added these details to the figure caption.

**12. Page 21, Figure 4: Please define in caption the size ranges that correspond to "coarse", "intermediate", and "fine".**

Response:

The size ranges have now been defined in both the figure caption and the main text.

**13. Page 25, Lines 657-660: This sentence is very difficult to understand please consider rephrasing. It would also be helpful to explain in more detail how probability density functions could be used to improve representations of INP in models.**

Response:

We have rewritten this section to make it clearer and to provide more detail of PDFs and their potential future use.

**14. Table S2: I found this table helpful for following the main text. I suggest moving it to the main text.**

Response:

We have moved this table to the main text as suggested.

**15. Table S3: The technique used to obtain the measurements should be indicated in the table. I assume that the OPC and SMPS were used to measure number concentration and calculate the surface area concentration, whereas the PM10 concentrations was determined gravimetrically from the filters? (If instead PM10 is calculated from the OPC and SMPS, then the details of the calculation should be provided, especially assumed particle density.)**

Response:

The PM10 concentrations shown in the table were taken from data obtained from the Israeli Ministry of Environment (IME) website station in Rehovot, located ~1 km from the sampling site, since this site was used previously by Reicher et al. (2018, 2019) and Gat et al. (2021). However, we also determined the PM10 concentrations using our OPC throughout the campaign assuming densities of both 1 g/cm3 and 2.65 g/cm3 to plot the time series data in Figure 2; the SMPS data was not used here since the instrument was not run continuously. This is now described in detail in Section 2.7 in the main paper, while the caption for Table S3 now includes details of where the data for each measurement came from.

**16. Figure S8: It seems like sometimes the data points used are entirely above the line that is supposed to indicate the average, which isn't possible mathematically. Please clarify.**

Response:

This is due to the inclusion of zeros in the average; sometimes, in one temperature bin, a droplet might freeze in that bin while no droplets will freeze in a separate blank run, but the zeros are not shown on a log plot. We have now added a comment in the figure caption stating that zeros are in included in the calculations but are not shown on the plot, hence the average being lower than the illustrated data.

**17. Figure S16: Why are only OPC data used in panel (a) whereas OPC and SMPS data are used in panel (b). Also, in panel (b) the SMPS data appear to be periodically unavailable, please explain.**

Response:

We have now included the SMPS data in panel (a). There are gaps in the SMPS as it was switched off every night, whereas the OPC was, apart from one time when it stopped measuring, operated continuously. We have now clarified this in the figure caption.

**18. Figure S17: It would be helpful if the air mass category ID was added to each trajectory.**

Response:

The category ID has now been added to each trajectory.

---

## Author Response (AR2)

We thank the Dr. Jie Chen and an anonymous reviewer for their comments and time. We have addressed all of Dr. Chen's comments and made changes to the manuscript as advised.

**Dr. Chen's report**

*"In the revised manuscript submitted by Tarn et al., the authors have addressed most of the questions posed by the reviewer. However, I have noted some minor comments that should be considered before its publication in Aerosol Research."*

*Minor comments:*

1. *This statement is confusing: "Size-resolved INP analysis has demonstrated that ice-nucleating activity during dust events increased with increasing aerosol particle size and concentration, and that the activity of supermicron particles was similar for different dust events, suggesting that common mineral species were controlling ice nucleation"*

   *What is meant by "ice nucleating activity"? If it refers to the ice nucleating activity of the dust aerosol, it would be affected by the concentration of aerosol particles. Conversely, if it refers to the ice nucleating activity of individual dust particles, it would not be influenced by concentration. Additionally, what specific chemical components are indicated by "common mineral species"?*

Response:

This alludes to the possibility that different parts of the ice nucleation community mean "activity" in different ways, e.g. some might mean it to be concentration. Here, the "ice-nucleating activity" refers to the number of ice active sites per unit [dimension] of material, typically per unit surface area ($n_s(T)$) or per unit mass ($n_m(T)$), i.e. the density of ice active sites together with the activation temperature of those sites (assuming the singular approximation). Normalisation of the INP concentration to surface area or mass provides a means of comparing INPs across materials and locations (where concentrations will vary). Therefore, INP concentration might not necessarily correlate with ice-nucleating activity (in terms of $n_s(T)$), but it did in the stated case. The "common mineral species" comment refers to materials that are common across the regions of interest (i.e. commonality) rather than being "typical" minerals (although this is also the case given that K-feldspar is expected to dominate the INP population given its presence in the region). We have reworded this slightly and hope that it is more clear:

> *"Size-resolved INP analysis has demonstrated that ice-nucleating activity during dust events increased with increasing aerosol particle size and concentration, and that the activity of supermicron particles was similar for different dust events, suggesting that mineral species common across these regions were controlling ice nucleation (Reicher et al., 2018; Reicher et al. 2019)."*

To try to make clear what we mean by "activity" in this case, we have also now added a definition of activity, as opposed to concentration, as being $n_s(T)$ throughout this paper by stating this in the first instance in which "activity" is mentioned in the Introduction.

2. *Figure 6: The parameterization for fertile soils from O'Sullivan et al. [2014] is not displayed in this Figure.*

Response:

We apologise for this oversight, O'Sullivan 2014 was shown in the original submitted version of the paper but a slightly different version of the figure was used in the revision and O'Sullivan was unfortunately missing. We have added the O'Sullivan 2014 fertile soil parameterisation back into the plot.

3. *Can authors mark the Fertile Crescent region in Figure 3 or other places of interest (e.g., black sea)? Alternatively, the authors may consider creating their own plot.*

Response:

We have modified Figure 3 to show land and water more clearly, and have marked important regions discussed in the text such as the relevant seas and the Nile Delta. We have also marked the approximate region of the ancient Fertile Crescent:

[Figure]

4. *Section 3.5: It seems that samples collected from clean and dust days have comparable ice nucleation activities. Does this suggest that dust particles do not significantly influence the ice nucleation activities of the observed aerosols?*

Response:

On the contrary, the fact that the active site density curves below ∼−18 °C are similar, despite the dust surface area varying substantially, is consistent with the dust controlling the INP population in this regime. As we say in the text, it is consistent with about 1 wt% K-feldspar, as per the Harrison et al. (2019) parameterisation.

5. *The authors claimed that samples have lower ice nucleation activities compared to fertile soils. Does this imply that the contribution of fertile soils is insignificant? Testa et al. [2021] also reported ice nucleating particles from agricultural emission, maybe add these results from comparison. How about other dust sources? Did authors consider other dust sources, for example, anthropogenic dust from cities as indicated in Chen et al. [2024].*

Response:

Testa et al. only provide their data for INP concentrations, which cannot be directly compared to our data given the different locations and environments. We have added the anthropogenic INP parameterisation from Chen et al. to Figure 1c and discuss it briefly in the text but given that this is also in terms of INP concentration, rather than ice-active site density (e.g. $n_s$(T)) (please see response to comment 1), and the environments between the two are very different, we can only suggest that anthropogenic dust could have an influence, although given the $n_s$(T) values discussed later in the manuscript we believe that the INP population is largely dominated by K-feldspar. We have reworded this addition thus:

> "*The recent parameterisation for anthropogenic INPs of Chen et al. (2024), which uses supermicron aerosol concentrations to estimate heat-resistant INP concentrations from sources*

*such as traffic-influenced road dust, sits at the top-end of our INP data below about −15 °C, though the sampling site, being in a ubiquitously dusty environment and prone to air masses from varied locations, was a very different to the metropolis from which the parameterisation was derived. It is therefore difficult to compare the parameterisation directly in terms of INP concentration, but given the presence of mineral dust in the region, particularly of K-feldspar, the influence of anthropogenic dust is not expected to be a dominant source of INPs."*

In the later discussion of $n_s(T)$, we have added parameterisations for Wyoming agricultural soil and China loess dust to Figure 6 and brief discussion that our results are lower than would be expected for such soils.